# Therapeutic Silencing of BCL-2 Using NK Cell-Derived Exosomes as a Novel Therapeutic Approach in Breast Cancer

**DOI:** 10.3390/cancers13102397

**Published:** 2021-05-15

**Authors:** Kübra Kaban, Clemens Hinterleitner, Yanjun Zhou, Emine Salva, Ayse Gülten Kantarci, Helmut R. Salih, Melanie Märklin

**Affiliations:** 1Clinical Collaboration Unit Translational Immunology, German Cancer Consortium (DKTK), Department of Internal Medicine, University Hospital Tübingen, 72076 Tübingen, Germany; yanjun.zhou@med.uni-tuebingen.de (Y.Z.); helmut.salih@med.uni-tuebingen.de (H.R.S.); 2Cluster of Excellence iFIT (EXC 2180) “Image-Guided and Functionally Instructed Tumor Therapies”, University of Tübingen, 72076 Tübingen, Germany; clemens.hinterleitner@med.uni-tuebingen.de; 3Department of Medical Oncology and Pulmonology, University Hospital Tübingen, 72076 Tübingen, Germany; 4Department of Pharmaceutical Biotechnology, Inonu University, 44280 Malatya, Turkey; emine.salva@inonu.edu.tr; 5Department of Pharmaceutical Biotechnology, Ege University, 35040 Izmir, Turkey; gulten.kantarci@ege.edu.tr

**Keywords:** NK cell, exosome, BCL-2, siRNA

## Abstract

**Simple Summary:**

Overexpression of the antiapoptotic protein BCL-2 is correlated with estrogen receptor (ER) expression in breast cancer and plays an important role for disease pathophysiology. Here, we conceptualized a novel treatment strategy by targeting ER^+^ breast cancer with NK cell-derived exosomes used as a carrier for BCL-2 targeted siRNAs. With this new approach, we successfully enhanced killing ability of NK cell derived exosomes by silencing of BCL-2 overexpression.

**Abstract:**

Overexpression of the anti-apoptotic protein BCL-2 is frequently observed in multiple malignancies, including about 85% of patients with estrogen receptor positive (ER^+^) breast cancer. Besides being studied as a prognostic marker, BCL-2 is investigated as a therapeutic target in ER^+^ breast cancer. Here, we introduce a new exosome-based strategy to target BCL-2 using genetically modified natural killer (NK) cells. The NK cell line NK92MI was lentivirally transduced to express and load BCL-2 siRNAs (siBCL-2) into exosomes (NKExos) and then evaluated for its potential to treat ER^+^ breast cancer. Transfected NK92MI cells produced substantial levels of BCL-2 siRNAs, without substantially affecting NK cell viability or effector function and led to loading of siBCL-2 in NKExos. Remarkably, targeting BCL-2 via siBCL-2 NKExos led to enhanced intrinsic apoptosis in breast cancer cells, without affecting non-malignant cells. Together, our prototypical results for BCL-2 in breast cancer provide proof of concept for a novel strategy to utilize NKExos as a natural delivery vector for siRNA targeting of oncogenes.

## 1. Introduction

Cancer globally is the second leading cause of death, and breast cancer is the most common cancer in women in industrialized countries [1,2,3]. Novel treatment options like improved surgery techniques (neoadjuvant) chemotherapeutics and immunotherapeutic strategies targeting the receptor tyrosine kinase Her2/neu have substantially improved survival of patients [4,5]. However, disease relapse and metastasizing disease remain a challenging problem, and multiple efforts are ongoing to develop new treatment strategies [5,6,7]. Among the most challenging obstacles for improved therapy, off-target side effects and multidrug resistance play a major role [4,8,9]. To address these needs, novel approaches to improve drug delivery systems and overcome treatment resistances are needed.

With 85% of the majority of estrogen receptor positive (ER^+^) breast cancer patients present with overexpression of the anti-apoptotic protein BCL-2 [3], which is an estrogen-responsive gene promoting tumor cell survival and disease progression by allowing breast cancer cells to evade programmed cell death [10,11,12,13]. Of note, BCL-2 inhibition via therapeutics like the BCL-2 inhibitor Venetoclax sensitizes cancer cells to apoptosis and has become indispensable for oncological treatment in Chronic Lymphocytic Leukemia (CLL) or Acute Myeloid Leukemia (AML) [14,15,16,17]. Besides targeting of BCL-2 with small molecule inhibitors, small interfering RNA (siRNA) is a powerful tool to posttranscriptional gene silencing [18]. However, therapeutic application of siRNA faces multiple obstacles like efficient delivery or activation of the complement system. Accordingly, suitable vectors to deliver the siRNA payload specifically to the tumor site are needed to overcome these issues [19,20].

Exosomes are homogeneous extracellular vesicles with around 30–150 nm in diameter and reportedly can be utilized as a natural nano-carrier [21]. They contain proteins such as Tetraspanins (CD9, CD63, and CD81) or Alix, representative for their exosomal origin as well as membrane ligands derived from their cells of origin which allow for communication with other cells [22]. Ligand–receptor interactions, fusion, and internalization via receptor-mediated endocytosis mediated by specific ligands govern the interaction of exosomes with their target cells. Recently, it was shown that exosomes contain cytosolic proteins of the host cell as well as non-coding RNAs [22,23]. They also can be modified, for example to express ligands which are specific for targeted cancer cells or loaded with active therapeutics [24]. Accordingly, they are suitable vehicles to overcome problems like, internalization into target cells, toxicity, and immunogenicity [25]. Previous work showed that exosomes naturally produced by NK cells (NKExos) can mediate antitumor effects by providing FasL and perforin/granzyme and are generally well suited to target malignant cells [26,27,28,29,30,31,32].

In this study, we here aimed to investigate if (i) NKExos could be used as delivery vector for siRNAs targeting oncoproteins and (ii) thereby enhancing their killing ability in solid tumors, (iii) with the final goal to induce the intrinsic apoptosis pathway in targeted tumor cells. To this end, we developed a therapeutic strategy to target overexpressed BCL-2 in ER^+^ breast cancer cells with exosomes loaded with small interfering RNA (siRNA) targeting BCL-2 (siBCL-2) derived from lentivirally modified NK cells.

## 2. Materials and Methods

### 2.1. Cell Lines and Primary Cells

The human NK cell line NK92MI was purchased from ATCC (Manassas, VA, USA). HEK293T, K562, MCF-7, MEC-1, SKBR3, MCF-10A, T-47D, and MDA-MB-231 cell lines were obtained from the German Collection of Microorganisms and Cell Cultures (Braunschweig, Germany). Receptor expression of the breast cancer cell lines is shown in Appendix A. Peripheral blood mononuclear cells (PBMC) of healthy volunteers were isolated by density gradient centrifugation (Biocoll; Biochrom, Berlin, Germany) and cultured in RPMI 1640 medium (LifeTech). T-47D cells were maintained in RPMI 1640 supplemented with 15% FBS, 10 µg/mL insulin, 100 μg/mL penicillin, and 100 mg/mL streptomycin. NK92MI and MEC-1 cells were cultured in IMDM medium (LifeTech, Paisley, UK). K562 cells were maintained in RPMI 1640 medium (LifeTech). HEK293T, MCF-7, MDA-MB-231 and SKBR3 cells were cultured in DMEM medium (LifeTech). MCF-10A cells were maintained in DMEM/F12 (LifeTech) supplemented with 5% horse serum, 20 ng/mL EGF, 0.5 µg/mL hydrocortisone, 100 ng/mL cholera toxin, 10 µg/mL insulin, 100 μg/mL penicillin, and 100 mg/mL streptomycin. Unless otherwise stated, each medium was supplemented with 10% FBS, 100 μg/mL penicillin, and 100 mg/mL streptomycin and cells are maintained at 37 °C with 5% CO_2_.

### 2.2. Lentiviral Vector Packaging, Titration, and Cell Transduction

HEK293T cells were transfected with envelope (pCMV-VSV-G), packaging (pCMV delta R8.2), and transfer vectors with TransIT (Mirus Bio, Madison, WI, USA) transfection reagent according to manufacturer’s protocol. Transfer vectors containing shRNA targeting BCL-2, GFP (pLKO.1), or control (Scrambled sequence, Scr) were used. The Stuffer-GFP expression vector was used as a GFP expression vector. The target sequences of the genes of interest are shown in Table 1. Virus-containing supernatant was harvested on days 2 and 3 after transfection, and viral titer was measured using qPCR Lentivirus Titer Kit (AbmGood, Richmond, BC, Canada). NK92MI cells were transduced with lentivirus (LV) at 20–100 multiplicity of infection (MOI) in the presence of Polybrene (8 µg/mL) (Sigma-Aldrich, St. Louis, MO, USA). Successfully transduced cells were selected by adding 1 µg/mL Puromycin (LifeTech) and controlling GFP expression by fluorescence microscopy.

### 2.3. Flow Cytometry

Cells were treated with siScr or siBCL-2 loaded NKExos at the indicated concentrations for 24 h. Intracellular flow cytometry using an anti-BCL-2 Ab (#15071, 1:100) (Cell Signaling, Danvers, MA, USA) followed by goat anti-mouse PE conjugate (1:100) (Agilent, Santa Clara, CA, USA) was performed using the Cytofix/Cytoperm Fixation/Permeabilization Solution Kit (BD Biosciences, Heidelberg, Germany) according to manufacturer’s instructions. Dead cells were excluded based on LIVE\DEAD™ Fixable Aqua (Invitrogen, Waltham, MA, USA) or 7-AAD (BioLegend, San Diego, CA, USA) positivity. TMRE (200 nM) (Sigma, St. Louis, USA) and Annexin V-PE (1:25) (BD Biosciences) were used according to manufacturers’ instructions. A FACS Canto (BD Biosciences) and FlowJo_V10.5.3 software (FlowJo LCC, BD, Ashland, OR, USA) were used for analysis.

### 2.4. Quantitative Reverse Transcription Polymerase Chain Reaction (qRT-PCR)

Total RNA was extracted using High Pure RNA Isolation Kit (Roche Diagnostics, Indianapolis, IN, USA) according to the manufacturer’s protocol. cDNA synthesis of 2 µg RNA was performed with 5× FastGene^®^ Scriptase II Ready Mix (Nippon Genetics, Dueren, Germany) and qRT-PCR was performed using 2× qPCRBIO SyGreen Mix no Rox (PCR Biosystems, London, UK) and QuantiTect Primer Assays (GAPDH; QT00079247 and BCL-2; QT00025011) according to manufacturers’ instructions.

### 2.5. Analysis of NK Cell Cytotoxicity

Lysis of K562 cells by NK92MI cells was assessed by 2 h Europium based cytotoxicity assays as previously described [36]. Specific lysis was calculated as follows:100 × (experimental release-spontaneous release)/(maximum release-spontaneous release)

### 2.6. Exosomes Isolation and Identification

To obtain NKExos, NK92MI cells were cultured in medium containing 10% exosome-depleted FBS (Biowest, Nuaillé, France) for 72 h. Cells and cellular debris were removed by centrifugation at 300× *g* for 10 min and at 16,000× *g* for 30 min at 4 °C. Next, exosomes were concentrated using 100K Amicon^®^ Ultra-15 centrifugal filters (Merck Millipore, Co cork, Ireland) and purified using Exo-spin™ Exosome Purification kit (Cell Guidance Systems, Cambridge, UK) according to manufacturer’s instructions. Purified exosomes were quantified using the CD63 Trific™ Exosome detection kit (Cell Guidance Systems) according to manufacturer’s instructions and stored at 80 °C until further use. Immunogold staining and Transmission Electron Microscopy (TEM) was performed by the DKFZ Core Facility Unit Electron Microscopy (Heidelberg, Germany). For immunostaining, CD56 (clone: MEM-188, Biolegend), CD63 (clone: H5C6, BD Biosciences), and isotype controls were used followed by a rabbit-anti-mouse antibody conjugated to 5 nm gold particles. The samples were analyzed using an EM910 (Zeiss, Oberkochen, Germany) transmission electron microscope at 63,000× magnification. Particle size of exosomes was determined by Nanoparticle Tracking Analysis (NTA) using a ZetaView^®^ Instrument (Particle Metrix, Inning am Ammersee, Germany) by Cell Guidance Systems.

### 2.7. Exosome Uptake

Exosome labeling was performed with a 1 µM Vybrant^®^ Dil cell-labeling solution (Invitrogen) followed by washing and ultrafiltration with 100K Amicon^®^ Ultra-15 centrifugal filters as previously described [37]. Dil-labeled exosomes were then co-cultured with GFP^+^ MCF-7 cells for 6 h. Nuclear staining was performed using NucBlue™ Live Cell Stain ReadyProbes™ reagent (Invitrogen) according to manufacturer’s protocol. After incubation, exosome transfected cells were washed with PBS and fixed in 4% paraformaldehyde (PFA) and analyzed by fluorescence microscopy. Image acquisition was performed using an Olympus BX63 microscope and a DP80 camera (Olympus). Exosome uptake was quantified by counting red and green fluorescent positive cells using Image J 1.53a software (National Institutes of Health, Bethesda, MD, USA).

### 2.8. Live Cell Analysis

Cells were treated with siScr or siBCL-2 NKExos, followed by addition of Incucyte^®^ Annexin V Red (1:200) or Incucyte^®^ Caspase-3/7 Red Dye (1:400) to the culture. For analysis of GFP silencing, GFP^+^ cells were treated with siScr or siGFP NKExos cells. Live cell imaging was performed with the Incucyte^®^ S3 Systems (Sartorius, Göttingen, Germany) for 48 h.

### 2.9. Caspase-9 Activation

Cells (1 × 10^4^ per well) were plated in a 96-well plate (NunclonTM Delta Surface, Thermo Fisher Scientific, Waltham, MA, USA) and incubated with the siScr, siBCL-2 NKExos or Staurosporin (2.5 µm) (Abcam, Cambridge, UK). After 24 h, Caspase 9 activity was measured using the CaspaseGlo^®^ 9 Assay (Promega, Madison, WI, USA) according to manufacturers’ instructions.

### 2.10. Statistical Analysis

Statistical analysis was performed with GraphPad Prism 8 (GraphPad Software, San Diego, CA, USA). Mean values with standard deviation (SD) are shown. The 95% confidence level was used, and normality distribution was determined by the Shapiro–Wilk test. *p*-values of normally distributed data were calculated by the multiple *t*-test or one-way ANOVA followed by Tukey’s multiple comparison test. In case of non-normality of the distribution, *p*-values were calculated by a Wilcoxon matched-pairs signed rank test or one-way ANOVA and a subsequent Dunn’s multiple comparison test.

## 3. Results

### 3.1. Generation of BCL-2 shRNA Producing NK92MI Cells

As a first step, we generated stable NK92MI transfectants expressing siRNA allowing for knocking down BCL-2 (NK92MI-siBCL-2) as well as scrambled controls (NK92MI-siScr) by lentiviral transduction as schematically depicted in Figure 1A. Successful transduction was assessed by measuring GFP expression with flow cytometry and fluorescence microscopy (Figure 1B,C). In line with gene expression analysis, which revealed a significant reduction of BCL-2 mRNA levels in NK92MI siBCL-2 compared to NK92MI-siScr cells (Figure 1D), we observed significantly reduced BCL-2 protein levels measured by flow cytometric analysis (Figure 1E,F). To further ascertain the effect of BCL-2 reduction in NK92MI cells, we performed viability assays based on 7-AAD inclusion, which revealed that NK92MI-siBCL-2 cells exhibited only slightly lower viability than NK92MI-siScr cells (82% vs. 72%) (Figure 1G). We next determined if BCL-2 downregulation in NK92MI cells affected reactivity against target cells using the human leukemia cell line K562. Cytotoxicity assays revealed no relevant effect of the BCL-2 siRNA on the ability of NK92MI cells to lyse target cells (Figure 1H,I). In conclusion, the altered BCL-2 level did not affect functionality of NK92MI cells, which provides a valid setting for further experiments regardless of the slightly decreased cell viability.

### 3.2. Characterization of NK Cell-Derived Exosomes (NKExos)

For production of exosomes, NK92MI-siBCL-2 and NK92MI-siScr cells were cultured for 72 h followed by collection of supernatants for exosome isolation. Purification steps of NKExos are depicted in Figure 2A and described in the Methods section. After purification, exosomes were investigated by Nanoparticle Tracking Analysis (NTA) ZetaView^®^, which revealed that the mean of size for siScr and siBCL-2 NKExos ranged between 115.8 128.9 nm (Figure 2B). The exosomes were additionally characterized by transmission electron microscopy (TEM) to determine whether size and shape of exosomes were in the expected range (Figure 2C). Phenotypic characterization was performed using immunogold staining for surface markers specific for exosomes (CD63) and NK cells (CD56) (Figure 2C), confirming that the isolated nanoparticles express respective characteristics. The concentration of the purified exosomes was investigated by CD63 specific TRIFic™ exosome detection assays and ranged between 467–1107 µg/mL and 450–890 µg/mL for siScr and siBCL-2 NKExos, respectively (Figure 2D). To investigate whether our modified exosomes were internalized by breast cancer cells, exosomes were labeled with the red fluorescent dye Dil and incubated with MCF-7 breast cancer cells. Exosome uptake was visualized by fluorescent microscopy after 6 h, which revealed Dil-positivity for 33% of the cancer cells, thereby confirming successful uptake of exosomes into cytoplasmic compartments (Figure 2E,F).

### 3.3. Silencing of GFP by siGFP Loaded NKExos

To test whether siRNA could be effectively loaded into NKExos and if the latter then would be capable of silencing the target gene of interest, we developed a GFP silencing system. We first generated GFP expressing breast cancer cells and transduced these GFP^+^ cells with lentivirus (LV) carrying shGFP (shGFP-LV) or shScr (shScr-LV) as control (Figure 3A and Appendix A). Subsequently, GFP expression was analyzed by flow cytometry which revealed that mean GFP expression of MCF-7 and MDA-MB-231 cells was decreased by 88% and 76% compared to shScr-LV, respectively. Next, we analyzed whether siGFP NKExos are capable of silencing GFP expression in MCF-7 and MDA-MB-231 cells to a comparable extent. Flow cytometric analysis revealed potent GFP silencing effects by siGFP NKExos (Figure 3B) and live cell imaging confirmed GFP loss to about 50% in the targeted cells (Figure 3C,D and Appendix A). These results demonstrated that siRNAs were naturally loaded into NKExos, and the latter represents an ideal siRNA delivery vector.

### 3.4. siBCL-2 NKExos Promote Apoptosis of Breast Cancer Cells

Several studies previously demonstrated that NKExos can induce apoptosis of tumor cells via transferring exosomal contents like FasL and perforin/granzyme into the tumor cells [28,29,30,31,32,38]. Furthermore, natural cytotoxicity of NKExos has been shown to be more effective on hematologic tumors cells compared to solid tumors, while healthy cells are less to none affected [32,38]. We determined whether and how treatment of different cancer cells with siBCL-2 NKExos is capable of inducing apoptosis. In addition to PBMCs from healthy donors, we utilized the CLL cell line MEC-1 in order to confirm the previous observed effect on hematological malignancy as well as the mammary cell lines: MCF-7, T-47D, SKBR3, MDA-MB-231, and MCF-10A. Cells were treated with siScr or siBCL-2 NKExos and induction of apoptosis was assessed by flow cytometry using Annexin V and 7-AAD staining as well as live cell imaging for Annexin V and Caspase 3/7 activation over time. Treatment of the non-malignant mammary cell line MCF-10A as well as PBMCs with siScr or siBCL-2 NKExos did not induce killing of these cells at all (Figure 4A,B). Analyses of MEC-1, MCF-7, T-47D, SKBR3 and MDA-MB-231 after treatment with NKExos showed enhanced killing capacity of siBCL-2 compared to siScr NKExos in ER^+^ MCF-7 and T-47D cells as well as the MEC-1 CLL cells (Appendix A), while MDA-MB-231 and to a lower extent SKBR3 showed Annexin V induction after NKExos treatment, but no differences between siScr and siBCL-2 NKExos (Appendix A). Of note, susceptibility to siBCL-2 NKExos correlated with the BCL-2 expression is assessed by qRT-PCR (Appendix A).

We found that more than 2-fold enhanced Annexin V positivity with siBCL-2 NKExos compared to siScr occurred in MCF-7 cells after 24 h, whereas earlier time points showed no effects (Figure 4C,D). Live cell imaging of MCF-7 cells revealed that Annexin V positivity increased exponentially upon siBCL-2 NKExos treatment, whereas siScr exposure had no effect (Figure 4E–G). Further inline, potent Caspase 3/7 activation was observed with siBCL-2 NKExos as compared to siScr over time (Figure 4H–J). Differences in Caspase 3/7 activation with MCF-7 cells reached statistical significance after 24 h, while no significance was observed at the earlier time points (Figure 4J).

### 3.5. siBCL-2 NKExos Enhance the Intrinsic Apoptosis Pathway of Breast Cancer Cells

Finally, we aim to confirm that the efficacy of siBCL-2 NKExos was indeed caused by silencing of BCL-2 and subsequent induction of intrinsic apoptosis. As a first step, we treated breast cancer cells with different concentrations of siBCL-2 and siScr NKExos as controls. Quantitative PCR analysis revealed a concentration dependent decrease in the BCL-2 expression after 24 h and treatment with 200 µg/mL siBCL-2 NKExos silenced BCL-2 mRNA expression about 50% after 48 h, although high concentrations of siScr NKExos did not affect BCL-2 mRNA levels (Figure 5A,B). Intracellular staining of BCL-2 protein by flow cytometry revealed a significant reduction in BCL-2 levels in breast cancer cells likewise upon treatment with 200 µg/mL of siBCL-2 NKExos (Figure 5C). Next, we analyzed whether the intrinsic apoptosis pathway was provoked upon siBCL-2 NKExos treatment. We found by TMRE staining that TMRE^−^ cells were increased by ~20% (Figure 5D, left) and mitochondrial membrane integrity showed a tendency to impair in siBCL-2 NKExos treated cells compared to siScr NKExos (Figure 5D, right). As TMRE^−^ cells indicate dysfunctional mitochondria associated with release of Cytochrome C and Caspase 9 activation, we analyzed whether siBCL-2 NKExos treatment disrupts mitochondrial potential and thereby induces Cytochrome C release. Analysis of Caspase 9 activity revealed an increase of about 45% upon siBCL-2 NKExos treatment compared to siScr NKExos (Figure 5E). Thus, in addition to cytotoxic effects of NKExos, we successfully demonstrated that mitochondrial-dependent apoptosis was activated by siBCL-2 NKExos as well.

## 4. Discussion

Breast cancer is the most common cancer in women [39]. Approaches to improve outcome include, beyond novel conventional systemic treatment compounds, immunotherapeutic approaches aiming to stimulate the reactivity of cytotoxic lymphocytes against the malignant cells [40]. Present strategies comprise checkpoint inhibition, Ref. [41] transfection with chimeric antigen receptors (CARs) [42] or application of bispecific antibodies to mobilize T cells, components of the adaptive immune system [43,44], against cancer. Stimulation of NK cells as cytotoxic lymphocytes of the innate immune system constitute an alternative to the promise, which is exemplified by their contribution to the efficacy of Her2/neu antibody Trastuzumab, where they mediate potent antibody-dependent cellular cytotoxicity (ADCC) [45].

Cancer progression per se is not only dependent on uncontrolled tumor cell proliferation, but also on reduced susceptibility of the malignant cells to apoptosis, which is deregulated by various mechanisms including upregulation of anti-apoptotic proteins like BCL-2 [11,46]. The relevance of BCL-2 for disease pathophysiology and also susceptibility to treatment is clearly documented for other malignancies such as CLL or AML, where this mechanism is successfully targeted using therapeutics like the BCL-2 inhibitor Venetoclax [16,17]. In breast cancer, BCL-2 likewise plays an important role in preventing apoptosis and upregulated expression correlates with estrogen receptor (ER) positivity [3,10,11,13]. Furthermore, it is known that BCL-2 overexpression is associated with an aggressive tumor phenotype and poor survival [10,47]. For this reason, we developed a siRNA-based gene silencing system using NKExos which could be used as a delivery vector for siRNAs targeting oncoproteins like BCL-2. One of the advantages of NKExos is their tumor specificity without affecting heathy cells [32,38]. Compared to NK cells, NKExos are skilled with the same beneficial effects like NK cells, tumor specificity, lysis of tumor cells by perforin/granzme, or induction of apoptosis via FasS/FasL. However, they have more advantages, since they are uptaken by the tumor cells and deliver exosomal contents like siRNAs, which could improve the targeted therapy. This is of great importance since tumor cells very often develop resistances against perforin/granzyme or Fas/FasL induced killing, or inhibit NK cell reactivity by expression of inhibitory molecules (e.g., MHC I). Moreover, heterogeneous antigen expression, the immunosuppressive tumor microenvironment, and natural barriers preventing penetration of the immune cells complicate the therapy of solid tumors [48]. Exosomes would be less affected by these mechanisms, since they deliver their cytotoxic contents as well as siRNA by membrane fusion or receptor mediated entry [49]. Our results demonstrated that NKExos loaded with siBCL-2 exhibited a promising killing ability in ER^+^ breast cancer and led to induction of Annexin V, Caspase 3/7, and Caspase 9 over time, which clearly demonstrates the efficacy of siRNA NKExos for treatment of breast cancer.

The impact of targeting BCL-2 in ER^+^ breast cancer has been evaluated in clinical phase I and II studies (NCT03900884 & NCT03584009) investigating BCL-2 inhibition by Venetoclax in combination with aromatase/cyclin-dependent kinase inhibitors and hormone therapy, respectively [50]. While demonstrating notable efficacy in these studies, in general, systemic BCL-2 inhibition caused lymphopenia, neutropenia, and thrombocytopenia, which required Venetoclax discontinuation in a substantial proportion of patients [51]. This led us to reason that strategies for specific targeting of cancer cells would constitute promising approaches to implement BCL-2 inhibition in ER^+^ breast cancer.

Specific targeting of BCL-2 by siRNA in various malignancies including breast cancer demonstrated growth inhibition, apoptosis induction, and enhanced susceptibility to chemotherapy in vitro and preliminary in vivo studies [12,52,53,54,55,56,57,58]. So far, clinical application of siRNA-based therapies is hampered by unspecific effects such as complement activation and induction of cytokine release. To overcome these issues, approaches like introduction of chemical modifications or loading siRNA into delivery vectors have been developed [19,59]. Previously, we used chitosan as a delivery vector to repress posttranscriptional vascular endothelial growth factor (VEGF) and zinc finger E-box binding homeobox (ZEB) expression, which resulted in inhibition of tumor angiogenesis and epithelial-mesenchymal transition (EMT) of breast cancer cells [60,61,62]. Even though chitosan has low cytotoxic, low bio-persistence, and mucoadhesive properties, weakness in intracellular dissociation and tumor cell specific targeting are still challenging problems, which reflects the necessity for improved delivery systems [63].

siRNAs formulated as a lipid complex or covalently linked to specific ligands enabled FDA approval of ONPATTRO^®^ (patisiran) for treatment of polyneuropathy in patients with hereditary transthyretin-mediated amyloidosis and GIVLAARI™ (givosiran) used for treatment of acute hepatic porphyria (AHP) [64,65]. However, developing specific delivery of siRNAs to malignant cells is greatly needed. Exosomes which deliver RNAs endogenously can easily be adapted to serve as optimal siRNA carriers [66]. Compared to the other gene delivery vectors, exosomes have the advantage that they can be actively loaded with therapeutics for delivery to the target sites and they are stable in the peripheral blood [67,68,69,70,71].

This holds also true for NK cell-derived NKExos, which besides the other advantages additionally express effector molecules like perforin/granzyme and FasL, per se enables killing of cancer cells exhibiting Fas receptors and being sensitive to apoptosis [26,27,28,29,30,31,32,38,72,73]. In general, natural cytotoxicity of NKExos has been shown to be less effective on solid tumors compared to hematologic tumor cells [38]. Moreover, in line with our results, NKExos showed no cytotoxic effects on non-malignant cells or PBMCs of healthy donors, which supports the tumor specificity of NKExos [32,38]. For this reason, we aimed to take advantage of the general antitumor effects of NKExos and combine this with their ability to serve as vehicles for delivery of siRNA targeting BCL-2 to enhance apoptosis in targeted solid tumor cells. Furthermore, we demonstrated that the efficacy of siBCL-2 NKExos is directly dependent on the BCL-2 expression in breast cancer cells. This suggests this new approach as particularly promising for ER^+^ breast cancer, as this subtype in particular is associated with strong BCL-2 overexpression.

## 5. Conclusions

In summary, we showed here for the first time that specific siRNAs can be lentivirally introduced in NK cells, which are then naturally loaded into exosomes and thereby increase apoptosis induction capacity on breast cancer cells. Furthermore, lentiviral modified NK92MI cells constitute an ideal tool for production of NKExos as a delivery vector for the targeting of specific oncoproteins in tumor cells, without affecting healthy cells. Overall, our study first provides evidence that siBCL-2 NKExos might serve as a powerful new tool for cancer therapy.

## Figures and Tables

**Figure 1 cancers-13-02397-f001:**
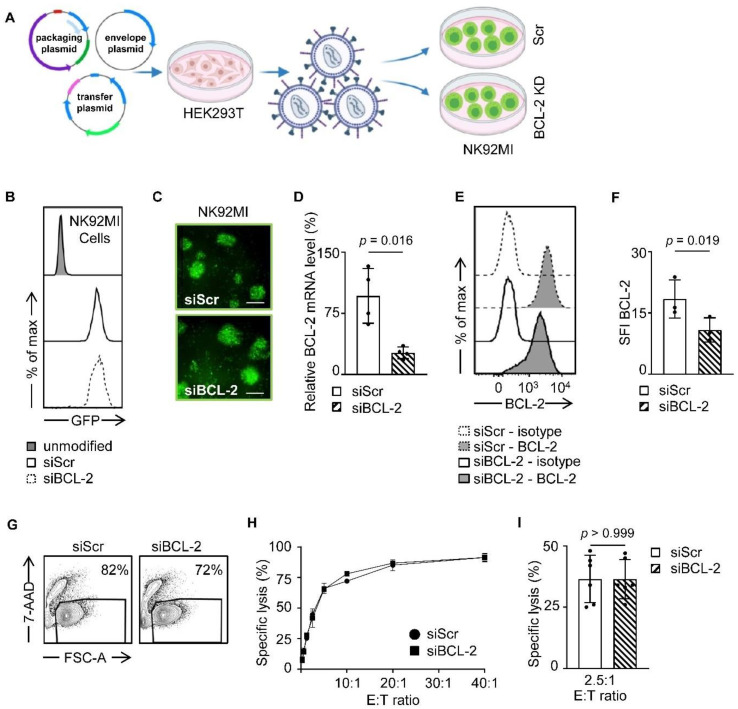
Generation of endogenous siBCL-2 producing NK cells. (**A**) Schematic workflow of lentiviral NK92MI transduction is depicted. HEK293T cells were transfected with different vectors, and LV-containing supernatant was used for transduction of NK92MI cells. (**B**,**C**) GFP expression levels of NK92MI-siBCL-2 and NK92MI-siScr cells were shown by flow cytometry and fluorescent microscope (Scale bar, 100 µm.), respectively. (**D**) Relative BCL-2 gene expression normalized to GAPDH of NK92MI-siBCL-2 and NK92MI-siScr cells is shown (*n* = 4). (**E**) Intracellular staining of NK92MI transfectants with anti-BCL-2 antibodies or isotype control was analyzed using flow cytometry. (**F**) BCL-2 expression level of NK92MI-siBCL-2 and NK92MI-siScr cells (*n* = 3) is shown as specific fluorescence indices (SFI) levels calculated by using median fluorescence intensity (MFI) of BCL-2/MFI Isotype control. (**G**) NK92MI transfectants were analyzed for 7-AAD positivity by flow cytometry. Living cell population is shown by exclusion of 7-AAD^+^ dead cells. (**H**,**I**) NK92MI transfectants were incubated with K562 cells at a different effector to target (E:T) ratios. Lysis of K562 cells was determined by 2 h cytotoxicity assays. (**H**) Exemplary results and (**I**) combined data at an E:T ratio of 2.5:1 are shown.

**Figure 2 cancers-13-02397-f002:**
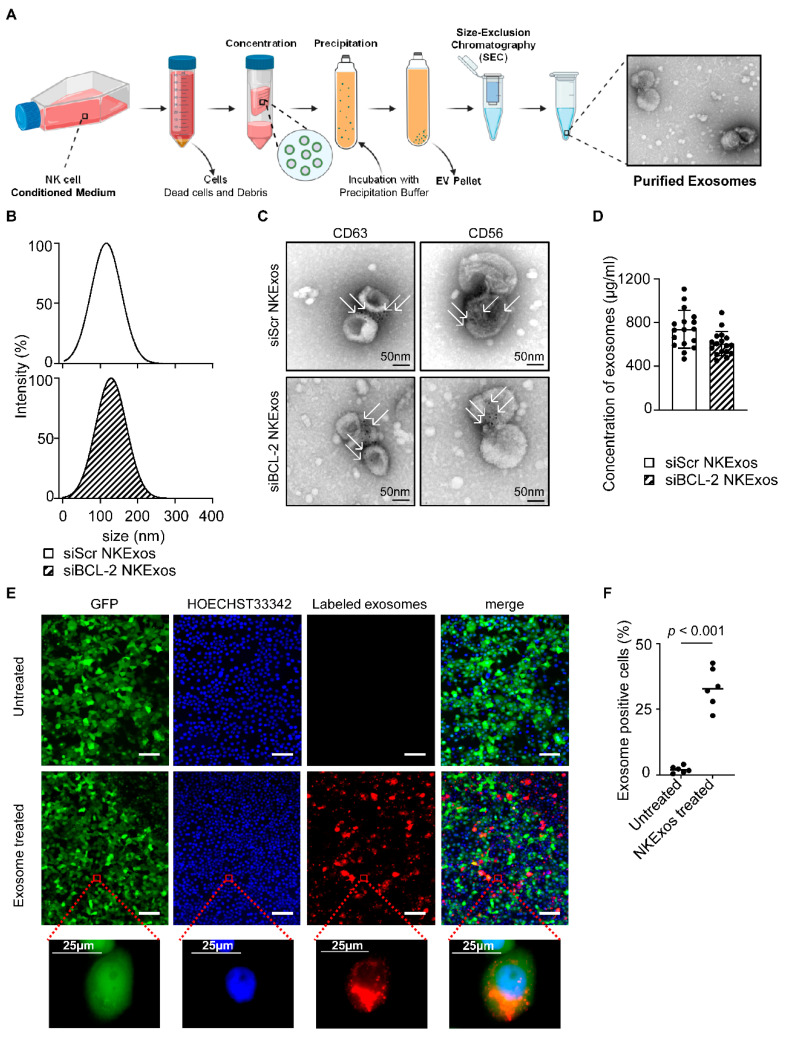
Isolation and characterization of NKExos. (**A**) Schematic workflow of NKExos production is depicted. NK92MI cells were cultured for 72 h, subsequently cells and cellular debris were removed by differential centrifugation and NKExos were concentrated and purified according to the method section. An exemplary TEM picture of isolated NKExos is shown. (**B**) Size distributions of siScr and siBCL-2 NKExos were measured by NTA showing peak diameters at 115.8 and 128.9 nm, respectively. (**C**) Immunogold staining of CD56 and CD63 on siScr and siBCL-2 NKExos was analyzed by TEM. Immunogold labeled exosomes with anti-CD63 and anti-CD56 antibodies are depicted (white arrows). (**D**) Exosome concentrations of siScr and siBCL-2 NKExos (*n* = 17) were calculated by Europium time-resolved immuno-fluorescence assay for detection of the exosome specific antigen CD63. (**E**) NKExos were labeled with fluorescent Dil and co-cultured with GFP^+^ MCF-7 for 6 h. Nuclear staining was performed using HOECHST33342 and NKExos internalization was analyzed by fluorescence microscopy (Scale bar, 100 µm). (**F**) Exosome uptake was calculated by counting double positive cells (*n* = 3) for GFP and red fluorescence.

**Figure 3 cancers-13-02397-f003:**
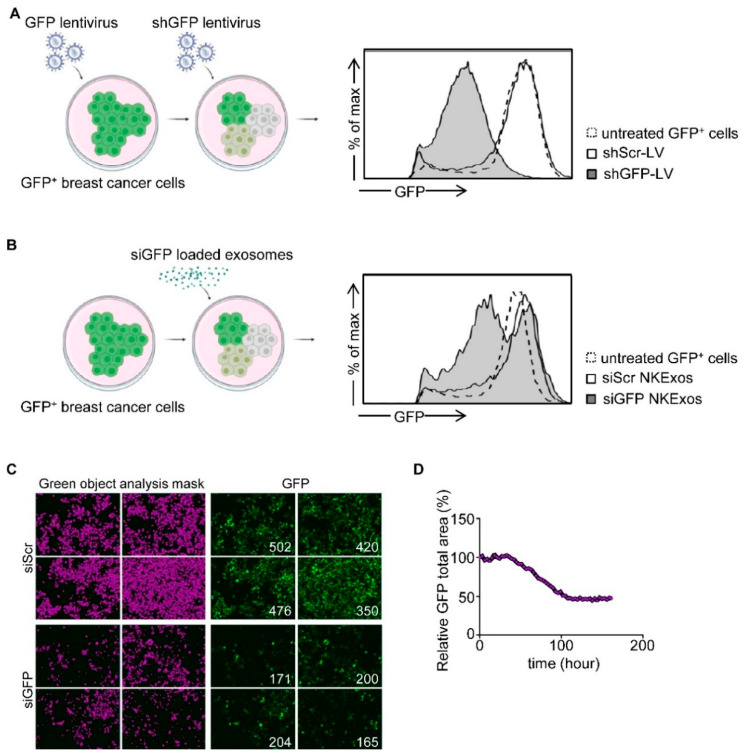
GFP silencing by siGFP NKExos. (**A**) Left, schematic workflow for LV transduction of GFP-expressing MCF-7 cells. Supernatants of HEK293T cells producing LV carrying shGFP or shScr were used to transduce GFP^+^ MCF-7 cells. Right, GFP^+^ MCF-7 cells were treated with shGFP and shScr LVs or untreated, and GFP expression levels were determined by flow cytometry. (**B**) GFP levels of GFP^+^ MCF-7 cells were analyzed after treatment with siGFP or siScr NKExos (200 µg/mL) for 48 h by flow cytometry. (**C**,**D**) GFP^+^ MCF-7 cells were treated with siGFP or siScr NKExos (200 µg/mL) over time, and GFP expression was analyzed with an Incucyte live cell imaging system. (**C**) Exemplary Incucyte pictures for green object mask analysis (purple, left) and GFP expression (right) is shown for siScr/siGFP NKExos treated MCF-7 cells after 72 h. Green object counts are depicted in the right corner of the pictures. (**D**) Relative GFP signal was calculated by dividing total GFP area of siGFP NKExos treated cells by total GFP area of siScr NKExos treated cells.

**Figure 4 cancers-13-02397-f004:**
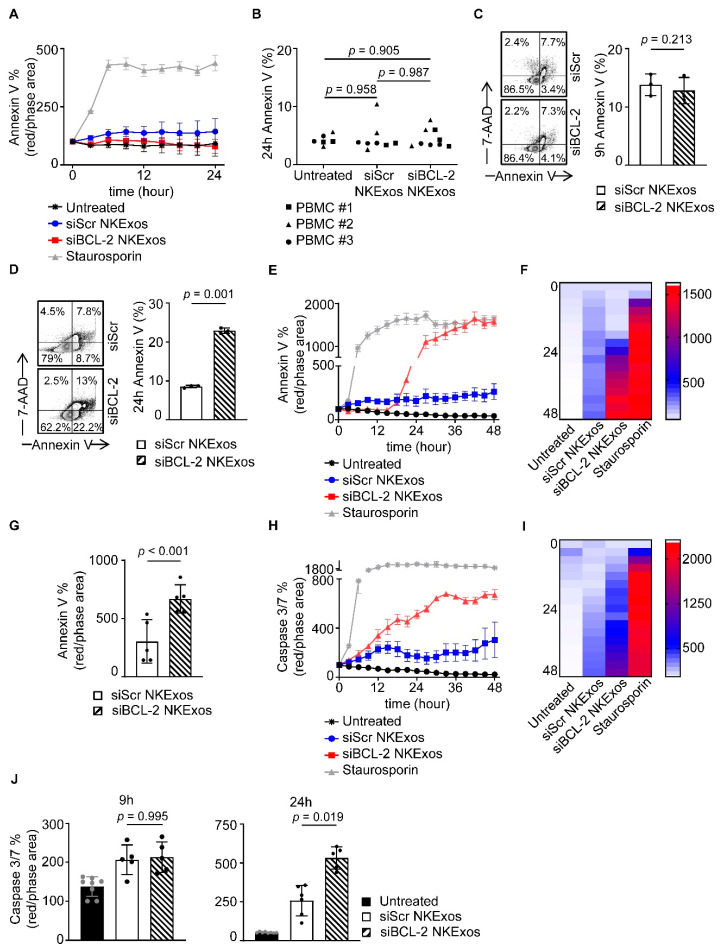
Apoptosis effects of siBCL-2 NKExos on malignant and non-malignant cells. (**A**) Annexin V expression after treatment with siBCL-2 or siScr NKExos (200 µg/mL) or staurosporine (2.5 µM) was determined by live cell imaging in MCF-10A cells, (**B**) PBMCs from three healthy donors were treated with siBCL-2 or siScr NKExos (200 µg/mL) and stained with Annexin V to detect apoptotic cells by flow cytometry. (**C**,**D**) MCF-7 cells were treated with siBCL-2 or siScr NKExos (200 µg/mL) and stained with Annexin V and 7-AAD to detect apoptotic cells. (**C**) Annexin V^+^ cells were analyzed after 9 h and (**D**) 24 h, respectively. (**E**–**G**) Annexin V expression after treatment with siBCL-2 or siScr NKExos (200 µg/mL) or staurosporine (2.5 µM) was determined by live cell imaging. (**E**) One exemplary result and (**F**) pooled independent experiments (*n* = 3) shown as heat map and (**G**) data achieved at 24 h are depicted. (**H**–**J**) Caspase 3/7 activation after treatment with siBCL-2 or siScr NKExos (200 µg/mL) or staurosporine (2.5 µM) was measured by live cell imaging. (**H**) An exemplary result and (**I**) pooled independent experiments (*n* = 3) were shown as heat map and (**J**) data achieved at 9 h (left) and 24 h (right) are depicted.

**Figure 5 cancers-13-02397-f005:**
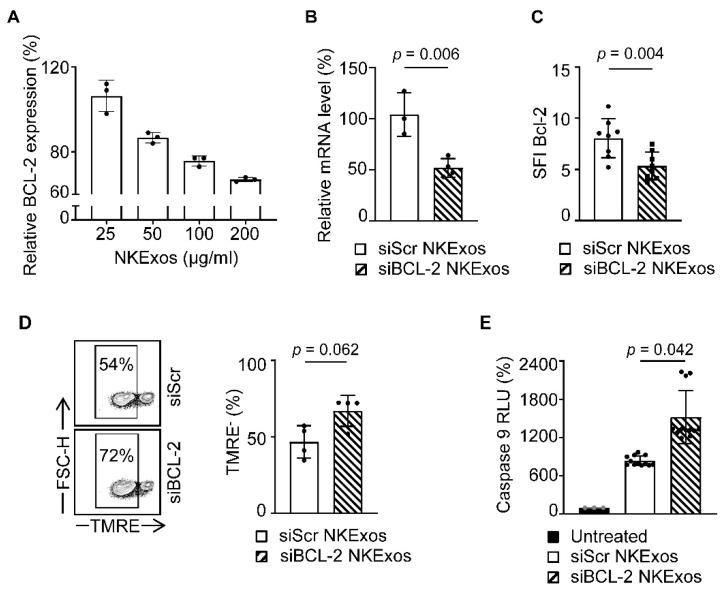
Analysis of the intrinsic apoptosis pathway after treatment of MCF-7 cells with siBCL-2 NKExos. (**A**) MCF-7 cells were treated with siBCL-2 or siScr NKExos at different concentration (25, 50, 100, and 200 µg/mL) and relative BCL-2 expression was determined by qRT-PCR after 24 h (relative BCL-2 expression of siBCL-2 NKExos treated cells was normalized to siScr NKExos treated counterparts). (**B**) BCL-2 mRNA levels were determined by qRT-PCR after treatment of MCF-7 cells with siBCL-2 or siScr NKExos (200 µg/mL) for 48 h (*n* = 3, Normalization was calculated according to BCL-2 mRNA levels of siScr NKExos treated cells). (**C**) SFI levels of intracellular BCL-2 expression assessed by flow cytometry in MCF-7 cells treated with siBCL-2 or siScr NKExos (200 µg/mL) for 24 h are shown (*n* = 4). (**D**) MCF-7 cells were treated with siBCL-2 or siScr NKExos (200 µg/mL) for 24 h and analyzed for TMRE staining by flow cytometry. Left, exemplary results and right, pooled data of three independent experiments are shown. (**E**) MCF-7 cells were treated with siBCL-2 or siScr NKExos (200 µg/mL) or left untreated for 24 h and subsequently analyzed for Caspase 9 activation. Pooled data for three independent experiments are shown.

**Table 1 cancers-13-02397-t001:** The sequences of genes of interest.

**shRNA**	**Reporter**	**Vector ID**	**Target Sequences (5′-3′)**
shScrambled	None	Addgene #1864 [33]	CCTAAGGTTAAGTCGCCCTCG
shGFP	None	Addgene #30323 [34]	GCAAGCTGACCCTGAAGTTCAT
shScrambled	eGFP	VB190131-1072gtc	CCTAAGGTTAAGTCGCCCTCG
shBCL-2	eGFP	VB190131-1073stf	CGGGAGATAGTGATGAAGTACATCCATTA
**Expression Vectors**	**Reporter**	**Vector ID**	
Stuffer	eGFP	VB181209-1075efx	
pCMV-dR8.2 dvpr	-	Addgene #8455 [35]	
pCMV-VSV-G	-	Addgene #8454 [35]	

## Data Availability

The data presented in this study are available on request from the corresponding author. The data are not publicly available due to privacy restrictions of voluntary donors.

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
