# Peer review of "Therapeutic Silencing of BCL-2 Using NK Cell-Derived Exosomes as a Novel Therapeutic Approach in Breast Cancer"

_cancers, 2021, doi:10.3390/cancers13102397_

Round 1

Reviewer 1 Report

In this manuscript, the authors use NK cells with siRNAs against BCL2 to produce siRNA-BCL2 carrying exosomes, which in turn are found to be taken up by breast cancer cells and inducing apoptosis. Overall, the manuscript is clearly written, but I do have some remarks.

My main concern is the rationale to use NK cells to isolate exosomes containing siRNAs. An important issue is what the specificity of the NK-cells and exosomes is: is there any tumor specificity, how are for example normal cells affected? Indeed, what is the advantage of using NK derived exosomes, compared to NK cells themselves, exosomes from other cells, other ways of targeting siRNA to tumor cells, or even the pharmacological approach using Venetoclax? None of this is addressed in this manuscript.

Somewhat related to this, it is not always clear why which cell lines were used. Why MEC1 cells? Why not show all data for MCF7 and MDAMB231 (or rather a number of different breast cancer cell lines, and possibly MCF10A) to show specificity for ER+, and/or tumor cells?

Further minor remarks:

Line 180: were the cells culture for 48 or 72 hrs?

Also show p-values when significant (not just **), and do not use 4 decimal values (2 or 3 are usually enough).

The authors apply multiple t-tests, but should rather use ANOVA in figs 4H/5E (also, in 4H/5E do not use black bars, individual data points are not discernable)

Figure 2E: higher magnification is necessary to conclude the exosomes are taken up. Has this uptake been quantified?

Line 274 right (not “rigth”) should be left, and left should be right. What do the numbers in fig 3C indicate?

Why use a heatmap to show 4 values? In fig 5A, relative BCL2 expression is >200% of values in the siBCL2 at 25 ug/ml compared to siScr? Please show actual values.

The qPCR data are not clear: what normalizer was used? Delta-delta Ct values (fig 5B): what compared to what? 10-5 is very low…

Reviewer 2 Report

The authors claim to discover a novel delivery system-NK cell derived exosomes, and therefore indicate that such mechanisms can be used as a potential therapeutic strategy. 

There are some feedbacks: 1. Could the authors delete the "Image are made with Biorender" or choose a paid version imaging process software?

2. I see the statistical analysis methods, however, according to the bar charts, many are very significant, but the p value are very low, could the authors double check that?

3. In order to claim this can be applied therapeutically, could the authors characterize after BCL-2 siRNAs, what processes lead to apoptotic phenomenon? The current analysis is a bit too general, using just Caspase3 and Annexin V. Some RT PCR can indicate gene regulation changes, perhaps. 

4. In the conclusion, did the authors accidentally included someone else's remark--"Authors should discuss the results and how 354 they can be interpreted from the perspective of previous studies and of the working hypotheses. The findings and 355 their implications should be discussed in the broadest context possible. Future research directions may also be 356 highlighted"? I also agree with the feedback. 

Reviewer 3 Report

The manuscript by Kaban et al examined therapeutic silencing of BCL-2 using NK cell-derived exosomes as novel therapeutic approach in breast cancer. This topic is interesting but not novel. In vivo study is missing. The translational potential of this method is unclear. The following major concerns need to be addressed before further consideration.

Major points:

  1. They need to validate the findings in vivo using breast cancer cell lines.
  2. They also need to use the PDX model to examine whether this method has the translational potential for patients.
  3. How to make sure the exosomes are uptaken by tumor cells rather than other normal cells. This is critical for in vivo study.
  4. In Fig. 5A, why would the siBCL-2 group have much higher expression of BCL-2 than that in siScr group when treated with 25ug/ml NKExos?
  5. They need to validate the alteration of apoptosis pathway as well as the expression of BCL-2 by western blot.
  6. The authors conceptualized a novel treatment strategy by targeting ER+ breast cancer with NK cell-20 derived exosomes used as carrier for BCL-2 targeted siRNAs. Is this treatment strategy only effective in ER+ breast cancer or can be applied to other subtypes of breast cancer? They should also test this treatment strategy in other subtypes.

Minor point:

  1. The figure legend of Fig. 3C does not match with the figure.

Reviewer 4 Report

In the manuscript titled “Therapeutic silencing of BCL-2 using NK cell-derived exosomes as novel therapeutic approach in breast cancer”, Kaban et al. used the NK cell secreted BCL-2 siRNAs-containing exosomes to silence the BCL-2 gene expression in breast cancer cell lines and induce apoptosis. Although the work is done with in vitro cell line models, these data provide a strong basis for the future testing of the efficacy of siBCL-2 exosomes in vivo. Overall, the study is well designed, with strong data to support all the claims. The manuscript is well written. 

My concern about this study is on the selection of the shBcl-2 sequence. It was not mentioned how the sequence was generated. Was it validated independently in a previous study? The length of the sequence seems to be unusually long. Was it a design or by chance? Also, if we change the target sequence of shBcl-2, how would the exosome work for silencing the BCL-2 gene in breast cancer cells?

Round 2

Reviewer 1 Report

The authors have done important additional experiments. Although the effect of treatment on the MCF10A cells is only followed for 24hrs, while the tumors cells are treated 48hrs, the data does convince that this approach is relatively specific, which greatly enhances the clinical relevance. As such, I have no further remarks.

Reviewer 3 Report

The authors have addressed all my concerns.